# Creative Performance and Conflict through the Lens of Humble Leadership: Testing a Moderated Mediation Model

**DOI:** 10.3390/bs13060483

**Published:** 2023-06-07

**Authors:** Haiou Liu, Syed Jameel Ahmed, Abdul Samad Kakar, Dilawar Khan Durrani

**Affiliations:** 1School of Economics and Management, Yanshan University, Qinhuangdao 066004, China; haiou@ysu.edu.cn; 2Department of Management sciences, University of Loralai, Loralai 84800, Pakistan; samad_kakar@uoli.edu.pk; 3Institute of Management Sciences, University of Balochistan, Quetta 83700, Pakistan; dilawar.commerce@um.uob.edu.pk

**Keywords:** humble leadership, creative performance, employee conflict, emotional intelligence, moderated mediation

## Abstract

This study developed and tested a moderated mediation model by examining the relationships between humble leadership (HL), emotional intelligence, employee conflict (EC), and creative performance (CP), using resource-based theory as the theoretical foundation. We conducted a cross-sectional survey of 322 employees and their immediate supervisors (*n* = 53) from the telecom sector in Pakistan. The data was analyzed using AMOS 21 and SPSS 26. The results demonstrate that HL has a positive effect on creative performance and a negative relationship with employee conflict. Furthermore, employee conflict has a negative impact on CP and mediates the impact of HL on CP. Moreover, a leader’s emotional intelligence moderates the negative relationship between HL and EC. Finally, this study reveals that EI moderates the indirect effects of HL on CP. The conclusions and implications are discussed at the end of this paper.

## 1. Introduction

In recent years, organizations have increasingly focused on employees’ innovative work behaviors, recognizing that these behaviors have a greater influence on organizational performance and can achieve better results [1]. However, insufficient focus has been given as to how employees can perform well [2] and achieve better performance [3] under the supervision of humble leadership (HL). This is particularly important given the potential for workforce diversity in organizations and differences in personalities and opinions [4]. Such differences among team members can lead to conflict [5], result in decreased productivity and performance, as well as waste organizational resources [6]. This is because employee conflict can lead to negative emotions, such as dissatisfaction, frustration, and irritation among team members, ultimately resulting in team dissolution [7] and poor organizational performance. Therefore, it is crucial for organizations to actively manage and resolve conflict among teams, while promoting diversity and inclusivity within the organization [8].

In the literature, various factors have been found to be essential in reducing employee conflict and enhancing creative performance (CP) [9]. One of the factors that is essential in resolving employee conflict and enhancing creative performance is humble leadership [10]. Humble leadership is defined as a leadership style that emphasizes sharing power, admitting mistakes, valuing input from others, and seeking to serve subordinates rather than being served by them [11]. This “bottom-up” leadership style is most successful in dynamic, uncertain, and unexpected contexts, where leaders face increasing challenges at the upper levels [6]. Research suggests that leaders who display humility may successfully minimize task conflict by honestly assessing themselves [8], treating others with appreciation, and welcoming new ideas with an open mind [12]. Scholars have also provided evidence of the impact of leadership on organizational performance [13]; however, limited attention has been given to the effects of humble leadership on creative performance [8]. Hence, this study’s first objective is to examine the impact of HL on employee conflict (EC) and creative performance.

Prior studies have suggested that HL’s effect on performance is not only direct; instead, other variables mediate it [14]. For instance, research has demonstrated that humble leadership enhances creative performance by fostering psychological safety [15], increasing knowledge sharing [16], and improving decision-making processes and collaboration among organizational members [17]. HL also promotes employees’ engagement and resilience, which can lead to increased job satisfaction and creative performance [18]. In addition, leaders who exhibit humility are more likely to create a positive work environment, which can reduce team conflict [6] and interpersonal conflicts within an organization [19]. This reduced conflict between the employees facilitated by HL, in turn, results in improved creative performance. However, empirical evidence of the impact of HL on CP via EC is limited [14]. Thus, this study’s second objective is to examine the mediating role of EC between HL and CP.

Furthermore, the resource-based (RBV) theory suggests that a firm’s performance depends on its resources [20]. These resources can take various forms, including material resources, such as financial and capital [21]; social resources, such as prestige and recognition and personal resources, such as skills and expertise [22]. This theory further suggests that a firm’s competitive advantage comes from its unique combination of resources and capabilities that are valuable, rare, inimitable, and non-substitutable [22]. Leaders’ emotional intelligence is one of the unique resources that is valuable to the company and can enhance the value of leader humility [23]. For instance, leaders with high emotional intelligence are more likely to display humility [24], which can enhance their perceived humility in the eyes of their subordinates. The increased humility of the leader is likely to increase the effect of humble leadership on EC [25]. In other words, this study suggests that EI is likely to moderate the indirect impact of HL on CP through EC. Figure 1 represents the hypothesized model of our study.

In sum, this study developed and tested a moderated mediation model, contributing to the literature in several ways. Firstly, it extends previous studies on humility by investigating how HL affects CP and EC. Secondly, this study delves deeper into the relationship between HL and CP by examining the role of conflict as a mediator. Finally, this study applies a resource-based view to explore the impact of the leader’s emotional intelligence (EI) as a contextual factor that may moderate the indirect effect of HL on CP through EC. 

## 2. Theoretical Background and Hypotheses

### 2.1. Humble Leadership and Creative Performance

As the modern workplace continues to evolve and shift towards more collaborative and team-based approaches, leadership styles have faced increasing scrutiny [2]. One leadership style that has gained traction in recent years is humble leadership [26]. Humble leadership is a style that emphasizes the importance of putting the needs and interests of team members first rather than prioritizing personal goals or agendas [27]. In essence, humble leadership is a people-focused approach that prioritizes creating an environment of trust and openness where subordinates feel comfortable speaking up and sharing ideas without fear of retribution or negative consequences [27]. This leadership style has been found effective not only in increasing individual and organizational performance but also in enhancing creative performance [28].

CP refers to an assessment of how effectively an employee is achieving his or her goals and objectives by performing creatively and generating and implementing new ideas in the workplace [29]. This assessment takes into account both the productivity and quality of the organizational output [30]; other factors, such as communication, collaboration, and problem-solving abilities, can also play an important role in enhancing the creative behavior of individuals [31]. A leader’s humility encourages pro-social behaviors and actions among subordinates, which eventually results in better CP [32]. 

Following RBV theory, we predict that HL is positively related to CP. This theory posits that a firm’s performance is dependent upon valuable resources. HL is an intangible organizational resource that inspires and motivates people and encourages individuals to perform [33]. As a result, motivated employees led by HL demonstrate improved CP [34]. Prior studies have also provided empirical evidence of the impact of HL on a firm’s performance. For instance, [35] found that HL is positively related to individual and creative performance. Humble leadership overcomes team members' weaknesses, which helps them to work more efficiently [36] and achieve the required task performance [37]. Some recent findings have also provided evidence of the impact of leadership on individuals’ CP and task performance [38,39,40,41]. Although these studies have extended our understanding of the impact of HL on individual performance, there is only limited knowledge of the relationship between HL and CP. Thus, based on the related literature and RBV theory, we hypothesize that:

**H1.** *Humble leadership will be positively associated with creative performance*.

### 2.2. Humble Leadership and Employee Conflict

Humility is considered a valuable resource that can build good social relationships through helpfulness. According to [42], humble individuals are more cooperative and are more likely to be seen as collaborative by their teammates. Humble leaders can serve as an inspiration and an ideal for followers, creating a peaceful and positive environment within the organization and fostering positive connections among the organizational members [37]. Following RBV theory, humility is also considered a resource of the organization [21]. Higher humility increases the value of other resources, such as negotiation and problem-solving skills, openness, and teaching ability [43]. These traits effectively manage social exchanges and decrease the negative effects of elements that cause stress and conflict [25]. Despite the importance of humility, the link between HL and EC has been neglected in the past.

Prior empirical findings have also demonstrated that humility can prevent or mitigate unwelcome behaviors that lead to employee conflict [25,44]. When subordinates witness their leaders acknowledging their weaknesses and maintaining modesty, they learn to follow suit by identifying and correcting their flaws [45]. Subordinates also learn to appreciate their colleagues’ contributions and demonstrate respect toward others, ultimately reducing employee conflicts [46]. As subordinates imitate the positive behaviors of their leaders, they develop mutual respect and tolerance and actively work towards resolving disputes that arise among the employees [47]. Therefore, a humble leader can facilitate productive interactions among individuals. Moreover, humble leadership fosters equitable decision making and power sharing, which promotes justice, openness, and fairness in teams [47]. This motivates employees to work towards common goals, fostering team cohesion and reducing conflict [5,6,22,48]. Thus, based on RBV theory and the related literature, we argue that:

**H2.** *Humble leadership is negatively associated with EC*.

### 2.3. Employee Conflict and Creative Performance

Conflict is a normal occurrence in personal and professional relationships, arising from disagreements and an inability to cooperate or understand each other’s limitations in the workplace [9]. Conflict may also arise from differences in work methods, task management, and personalities, leading to EC that can impede the organization’s development and performance [49]. Disagreements and clashes due to differences in opinions and aspirations can lead to EC [50], causing mental and physiological issues and affecting employees’ attitudes and job-related performance [46]. Conflict literature distinguishes between task and relationship conflicts, where task conflict may have positive, negative, or no significant consequences on creative performance [51], while relationship conflict is always dysfunctional [25]. 

Employees’ creative performance involves collaboration among individuals who share organizational tasks and a common goal to achieve the objectives [52], forming a distinct unit within the broader organizational structure. When workers interact to accomplish personal, organizational and team goals, they may encounter negative circumstances that affect their productivity and creativity [53], such as conflict in the workplace [54]. Research has shown that conflicts reduce employee performance and may cause workers to be isolated and engage in negative activities that further harm their innovative behavior [42]. De Dreu et al. [55] reported a negative correlation between task conflict and team effectiveness. Zhang and Huo [56] reported that project performance is negatively related to conflict among employees. We argue that conflict between individuals reduces their ability to focus on job-related activities. In other words, conflict drains cognitive resources that are needed for job-related performance [50] and thus reduces focus and lowers performance. RBV also explains the impact of employee conflict on CP. This theory suggests that a team’s resources, such as its collective knowledge, skills and abilities are negatively affected by conflict, ultimately leading to lower performance [57]. Based on RBV theory and the literature, we hypothesize the following:

**H3.** *Employee conflict will be negatively associated with creative performance*.

### 2.4. The Mediating Role of Employee Conflict

In this study, we argue that high levels of humility (HL) are related to creative performance (CP) not only directly but also indirectly through employee conflict (EC) [58] for the following reasons. First, research has shown that HL has a negative relationship with EC [59], and other studies have found that humility reduces employee conflict in challenging environments [48]. Additional evidence of the impact of HL on employee conflict has also been provided by [6], Li, Wei [60]. Second, studies have suggested that team members who are cooperative and free from conflict are more likely to perform better [13]. For example, Sackett [61] found that EC reduces CP, while the absence of conflict increases CP. Zhang and Huo [56] argued that employee conflict reduces performance, while Ye et al. [18] found a negative correlation between employee job satisfaction and job performance. In sum, these studies suggest that HL reduces employee conflict, which in turn enhances creative performance. On the other hand, RBV theory suggests that humble leadership can influence creative performance by reducing employee conflict [62], which can be considered a valuable intangible resource. By creating a cooperative and harmonious team environment, humble leaders can facilitate the development of valuable social capital within an organization [63]. This social capital can then be leveraged to create a sustainable competitive advantage by enabling employees to work together more effectively and efficiently, leading to better performance outcomes. Moreover, the reduction of employee conflict can also contribute to the development of a positive organizational culture [64], which can be a valuable resource for the firm’s long-term performance. Therefore, we posit that:

**H4.** *Employee conflict will mediate the positive relationship between humble leadership and creative performance*.

### 2.5. The Moderating Role of Emotional Intelligence

In this study, we predict that the impact of HL on EC is moderated by EI. In simple terms, leaders with high levels of EI may be better able to manage and reduce EC compared to those with low EI. EI is the ability to recognize, understand, and control emotions in oneself and others [65]. Individuals with high EI are better equipped to process and manage emotions, which can help them to resolve conflicts more effectively [25]. More specifically, HL can better manage EC if the leader has a high EI and vice versa. This is because leaders with high EI are better able to recognize and understand their subordinates’ emotions and respond in ways that diffuse or resolve conflicts positively [66]. They are also more likely to display positive attitudes and behaviors towards team members, which can help to strengthen relationships and build trust within the team as well as between individuals [67]. For instance, Schutte, Malouff [68] found that having high EI is associated with resolving challenging situations and accomplishing tasks. Additionally, those with a higher EI are more adept at resolving disagreements than people with a lower EI [67]. Mayer, CARUSO [69] found that individuals with high EI can reduce interpersonal conflict, and those with low conflict are more likely to perform well. 

We suggest that when leaders have higher levels of EI, the size and impact of the favorable association between HL and EC is greater. That is, highly emotionally intelligent HL is more likely to lead to more successful EC resolution than a lower emotionally intelligent HL, which ultimately influences creative performance. For instance, Schutte et al. [68] discovered and found that having high EI is associated and linked with resolving challenging situations and helps in accomplishing tasks. Additionally, those with higher EI are more adept at resolving disagreements than people with lower EI. Hence, we can hypothesize the following:

**H5.** *A humble leader’s EI moderates the negative relationship between the leader’s humility and employee conflict. The weaker the EI of a leader, the stronger the negative impact of the humble leader on employee conflict will be. The stronger the EI, the weaker the negative impact of humble leadership on employee conflict will be*.

### 2.6. The Moderated Mediation Effect

We also predict that EI moderates the indirect effect of HL on creative performance via employee conflict. In other words, this study suggests that humble leaders’ EI moderates the relationship between their humility and creative performance, and the mediating effect of employee conflict on this relationship is also influenced by the leader’s EI.

The ability of emotionally intelligent individuals to regulate their emotions enables them to think more rationally and identify genuine opportunities to resolve problems effectively [29]. Emotionally intelligent people are capable of finding solutions that benefit all parties involved in a conflict, thus promoting positive outcomes [56]. The emotional intelligence of a humble leader plays a crucial role in helping his or her members to manage conflicts and improve workplace efficiency by anticipating the negative emotions that may impede collaboration and creative performance [23]. Leaders who possess emotional intelligence have the ability to regulate their own emotions and those of others, which can help prevent disagreements from negatively affecting creative performance by defusing problematic situations and promoting positive relationships [70]. The development of humble leadership and emotional intelligence reduces the likelihood of undesirable emotions and approaches resulting from relational conflicts, which in turn results in improved creative performance [71]. As a result of the HL’s emotional intelligence, the CP is improved while disputes may be avoided. We accordingly propose the following hypothesis: 

**H6.** *The EI of a humble leader moderates the mediating effect of employee conflict on the link between the humility of a leader and creative performance. The weaker the EI of a leader, the stronger the mediating effect of employee conflict on the relationship between HL and creative performance. Conversely, the stronger the EI of the leader, the weaker the mediating effect of employee conflict on the relationship between HL and creative performance*.

## 3. Method

### 3.1. Sample and Data Collection Process

This study’s sample was selected through a convenient sampling method, targeting employees who were employed in the telecom sector in Pakistan. We selected the telecom sector because of the rapid technological changes in this sector. Due to these changes, employees are on their toes to incorporate these changes to meet the customer’s demands. This ever-changing working environment requires creativity and doing things in an innovative and creative way. Moreover, another reason for the selection of this sector is the lack of research in Pakistani, as well as in an international context. The data was collected from two provincial capitals of Pakistan, Quetta and Karachi. We collected data from these cities to increase the generalizability of our study, because the cities are different in culture, language, and working environment. To collect data, one of the researchers made several visits to the selected locations and obtained formal authorization from the CEOs and managers of the sites in order to enlist various employees and their direct supervisors. All the subjects gave their informed consent for inclusion before they participated in this study.

To ensure that the survey received the best and most accurate responses, researchers visited ten (10) offices in the telecom sector before formal data collection began to gather information on the education level, age, and working hours of the employees. Since we used convenient sampling, specific steps were taken to improve the representativeness of the sample. To minimize selection bias, we made efforts to select participants from various accessible locations and sources so that a diverse range of perspectives could be provided. It was ensured that a wide range of opinions and experiences were provided that represented the population as a whole. Moreover, we ensured the heterogeneity of the sample by rigorously seeking participants who had various demographic characteristics, such as age, gender, tenure, socio-economic status, etc. Finally, we applied rigorous data analysis techniques and carefully interpreted the results to reduce any biases due to convenient sampling. The data was collected in a 6-month time frame, from October 2021 to March 2022.

The initial survey revealed that the majority of the workforce had completed their bachelor’s degrees, but we found that the employees working in the lower grades were not highly qualified. Therefore, the researchers involved academicians from the linguistics department of the University of Balochistan, Quetta, Pakistan, for the translation of the survey questions into Urdu (the national language), using the traditional back-translation technique from English to ensure accuracy [72]. Thirty employees from three different telecommunications firms pre-tested the translated version, and feedback was collected on items that were unclear or challenging to comprehend. No major changes were made after this preliminary observation.

Initially, 500 surveys were distributed, including 53 to immediate supervisors or team leaders in the concerned sector. These companies (Ufon, Jaz, and Zong) include the three major shareholders of the market in telecommunication services to the public. Out of the 500 surveys, 344 were collected, which accounts for 69% of the total. However, 12 surveys were discarded due to missing responses or the marking of repetitive responses to different questions. Thus, 322 useful surveys remained for further analysis, which represents 64.4% of the total [73]. Required standards set criteria for the minimum sample size. According to this criteria, there should be five or less variables in a study, and there should be at least three measurement items. If a study meets the above-mentioned threshold, then the minimum sample size should be 100. Moreover, according to this criteria, the communalities should be greater than or equal to 0.6. Our sample size, number of variables, and communalities fulfills the required standards to justify our sample size.

All the respondents were categorized into six groups based on age. The majority of the respondents were between the ages of 20 and 40 years, with 87.6% of the total workforce being male and 12.4% being female. Additionally, 6.2% of the respondents had an intermediate schooling level, 20.4% had a higher secondary school certification, and 73.4% held a bachelor’s or master’s degree. The demographic data showed that only 3 workers were divorced or widowed, while 80.4% were married and 18.6% were unmarried.

### 3.2. Measures

#### 3.2.1. Leader’s Humility

The construct of humility was measured with a nine-item scale adopted from [30]. “This person actively seeks feedback, even if it is critical,” is one of the sample items from the adopted scale. The responses were marked on a five-point Likert scale ranging from (1) strongly disagree to (5) strongly agree. Cronbach’s alpha for the scale was 0.96.

#### 3.2.2. Creative Performance

We adopted the nine-item scale from Janssen [74] for measuring creative performance (CP). As an example, one item from the scale is “I create new ideas for difficult issues”. Managers were asked to evaluate the performance of their subordinates for each item. Cronbach’s alpha for the scale was found to be 0.97.

#### 3.2.3. Employee Conflict

A scale developed by Wright, Nixon [47] was adopted to measure workplace conflict among the workers. The scale includes items such as “Have you ever felt that you were treated unfairly by others at work?”, with response options ranging from strongly disagree (1) to strongly agree (5). Cronbach’s alpha coefficient for the scale was calculated to be 0.95, indicating high internal consistency and reliability.

#### 3.2.4. Emotional Intelligence

The EI of the leader was assessed using a 16-item scale developed by Wong and Law [75]. One of the items from the scale is “I really understand how I feel.” The EI scale used a 5-point Likert scale with response options ranging from strongly disagree (1) to strongly agree (5). The scale demonstrated high internal consistency and reliability with a Cronbach’s alpha coefficient of 0.98.

All the items from the measurement scales are mentioned in Appendix A.

### 3.3. Control Variables

Demographic factors were controlled for in this study, as past research has shown that they can have an impact on both leadership and team members [76,77]. Specifically, we controlled for firm size, industry type, experience, age, and gender to minimize their potential effects on this study [27,78].

## 4. Results

### 4.1. Reliability and Validity

The reliability of the scales used in this study was tested using SPSS 26.0 and AMOS 21.0, including confirmatory factor analysis (CFA). CFA analysis was performed to check the factor loadings, Cronbach’s alpha, and composite reliability for each construct. All the items were loaded against their construct and all the values of the factor loadings were above the minimum threshold of 0.70, as shown in Table 1. The results of Cronbach’s alpha indicated high levels of reliability, with alpha (α) values of 0.968 for HL, 0.976 for CP, 0.956 for EC, and 0.984 for EI. The values of CR also exceeded the minimum threshold of 0.70. After performing the preliminary analysis, we calculated convergent validity as evidenced by AVE, and all were above the minimum threshold of 0.5. The variables of this study also ensured discriminant validity, as the square root of AVE was greater than any inter-factor correlation.

The results of the correlation analysis, mean and standard deviation are present in Table 2. Consistent with our literature review, we found a positive relationship between HL and CP (r = 0.671, *p* < 0.01), indicating that the leader’s humility contributes to teamwork in an organization. We also found a negative relationship between EC and both HL (r = −0.599, *p* < 0.01) and CP (r = −0.597, *p* < 0.01). Additionally, we found a positive relationship between EI and both HL and CP, r = 0.395, *p* < 0.01 and r = 0.416, *p* < 0.01, respectively, while EI had a negative correlation with EC (r = −0.342, *p* < 0.01).

Table 3 shows a good model fit for the construct validity and the four factor models, as evidenced by the values of the model fit indices: ꭓ^2^/df = 2.75, comparative fit index (CFI = 0.937), root mean square error of approximation (RMSEA = 0.07), standardized root mean square residual (SRMR = 0.072), goodness-of-fit (GFI = 0.765), normal fit index (NFI 0.905), and the Tucker–Lewis Index (TLI = 0.929) [79].

### 4.2. Common Method Bias/Variance

To avoid potential common method bias, we conducted Herman’s single-factor test, which revealed that a single factor explained less than 50% of the variation, indicating no common method bias.

### 4.3. Hypotheses Testing

To test the hypotheses, PROCESS macro (Plug in for SPSS) and hierarchal regression analysis were used. The results show that HL had a total effect on CP (β = 0.652, *p* < 0.001, LLCI = 0.568, ULCI = 0.736) and EC (β = −0.562, *p* < 0.001, LLCI = −0.648, ULCI = −0.477), with a non-zero confidence interval, providing statistical support for Hypotheses 1 and 2, respectively (see Table 4). Additionally, the direct effect of EC on CP was also significant (β = −0.632, *p* < 0.001, LLCI = −0.724, ULCI = −0.541), supporting Hypothesis 3.

To test Hypothesis 4, the results in Table 4 show that the direct effect of HL on CP was partially mediated by EC (β = 0.4497, *p* < 0.001, LLCI = 0.3532, ULCI = 0.5463), as the β value was less than the value of the total effect. The indirect effect of EC (mediator) was also significant (β = 0.2019, *p* < 0.001, LLCI = 0.1382, ULCI = 0.2713), with a non-zero confidence interval, indicating that EC not only had a significant relationship to CP, but also a partial mediation effect or indirect effect on the relationship between HL and CP. Therefore, Hypothesis 4 was also supported (see Table 5).

The predictors in Model 1 explained the 34.2% variance in the outcome variable with F (2, 319) = 91.212, *p* < 0.001, with humble leadership (β = −0.527, *p* < 0.001) and emotional intelligence (β = −0.159, *p* < 0.01) predicting employee conflict. In Model 2, the predictors explained the 37.6% variance in the outcome variable with F (3, 318) = 63.860, *p* < 0.001, with humble leadership (β = −0.483, *p* < 0.001), emotional intelligence (β = −0.147, *p* < 0.01), and the HL × EI interaction (β = −0.121, *p* < 0.05) also predicting employee conflict. The ∆R^2^ value of 0.012 showed a 1.2% change in the variance of Model 1 and Model 2, with the difference between their F values being statistically significant, ΔF (1, 318) = 6.190, *p* < 0.05. Therefore, the findings suggest that emotional intelligence has a moderating effect on the negative relationship between humble leadership and employee conflict.

### 4.4. Moderation Graph

To minimize the risk of multi-collinearity, we standardized the independent variables (IV) HL and EI (moderator), and obtained their interaction term HL*EI [80]. The results in Table 6 show that the interaction item had a significant effect on EC (β = 0.121, *p* < 0.05), indicating that a humble leader’s EI moderates or alters the relationship between HL and EC. Figure 2 further supports this finding, as it displays the common interaction point, where a higher level of emotional intelligence dampens the negative relationship between humble leadership and employee conflict. Therefore, we accept H5 and conclude that a humble leader’s EI moderates the negative impact of HL on EC. The stronger the EI, the weaker the negative impact of humble leadership on team conflict is, and the weaker the EI, the stronger the negative impact of humble leadership on employee conflict is. The summary of the findings is highlighted in Figure 3. 

To analyze the moderated mediation effect, we used the model proposed by Hayes [81]. We utilized the PROCESS macro of SPSS to obtain the conditional indirect effect of emotional intelligence as a moderator. Table 6 shows that when EI was added to the mean value, the indirect effect was 0.1276, with a non-zero confidence interval of [0.0736, 0.1892]. This indicates that humble leadership has a significant indirect effect on creative performance through employee conflict. Similarly, when the mean value was reduced by one standard deviation, the indirect effect was 0.2184 with a non-zero confidence interval of [0.1316, 0.3174], indicating a significant indirect effect of humble leadership on creative performance through employee conflict.

Subsequently, the moderated mediation effect was obtained using the same PROCESS operation as indicated by INDEX in Table 7. The index indicates the indirect effect of the moderator at two different levels of parameter estimates, testing whether the moderator changes the mediating effect. The value of the moderating effect of emotional intelligence on the indirect relationship between humble leadership and creative performance was −0.328, with a 95% confidence interval of [−0.0688, −0.0048], which does not contain 0, indicating a significant moderated mediation effect. Therefore, we accept H6, suggesting that the emotional intelligence of a humble leader does moderate the indirect relationship between humble leadership and creative performance through employee conflict.

## 5. Discussion and Conclusions

Drawing on resource-based view theory, this study investigated the impact of humble leadership and emotional intelligence on employee conflict and creative performance in the telecom sector. The findings of the collected data from two cities in Pakistan indicated that a humble leader is positively associated with creative performance and negatively associated with employee conflict. Furthermore, employee conflict negatively predicted employee creative performance. To summarize our findings, we can say that HL can help employees to perform better and in innovative ways and can reduce conflict among employees. Moreover, we also found that emotional intelligence moderates the relationship between humble leadership and employee conflict. Additionally, we examined emotional intelligence as a potential moderator of the indirect relationship between humble leadership and creative performance through employee conflict; our findings and results show that emotional intelligence does influence this relationship. In conclusion, our study highlights that emotional intelligence plays an important role in the relationship between humble leadership, employee conflict, and creative performance, and that leaders with higher emotional intelligence can enhance the impact of humble leadership on creative performance through their effective management of employee conflict.

### 5.1. Theoretical Implications

This study highlights the positive impact of humble leadership (HL) on creative performance (CP), specifically in the telecom sector where rapid technological changes have increased the challenges many fold. The chances of the survival of an organization will be higher if it adopts changes and develops innovative solutions for existing problems. Previous studies on HL have mainly focused on its effectiveness at the project or team level [82], with very few exploring its impact at the organizational level, and even less in the telecom sector. Although Ou, Tsui [30] analyzed the impact of HL on corporate performance in the information technology industry, no prior studies have investigated the impact of HL on CP in the telecommunications sector, specifically in a Pakistani context. Therefore, this empirical study examines the role of HL on employee conflict (EC) and CP, contributing to the effectiveness of HL from the individual to the organizational level.

Furthermore, our study opens a new arena for investigating the transmission mechanism of the effectiveness of HL on CP, and discussing humble leadership’s impact on employee innovative behavior in organizations through employee conflict. Owens and Hekman [6] found that humble leadership can positively affect performance and promote a harmonious culture at the collective level, and that employee engagement is advantageous for both humble leaders and top management to incorporate as strategies, as both the team and its leader are integral parts of the organization’s internal environment. However, the relational role of humble leadership in effecting creative performance has largely been neglected.

Our study also explores employee conflict as a potential mediator of the relationship between humble leadership and creative performance. It highlights the importance of synergy through the use of HL on the organizational level. Good relationships between organizational members promote synergy, giving creative performance an edge over the old-fashioned routine work. However, if internal conflicts among employees are not handled properly and in a timely manner, they may escalate and result in turnover and decreased performance [83]. Therefore, this study reveals not only the team’s rational thinking process, but also the importance of managing employee conflict to enhance the effectiveness of HL on CP.

### 5.2. Practical Implications

In most organizations, from top to bottom, the typical authoritative style of leadership is followed. This is especially true in countries such as Pakistan, where people associate a leader with a high level of power and authority. However, in today’s dynamic conditions of business and technology, the typical and old-fashioned leadership style should be avoided, or at least be least preferred.

The uncertainty in every sector of industry and business has increased many fold since the outbreak of the COVID-19 pandemic. It has changed the dynamics of the working environment, making it quite difficult for top management to make policies and dictate them to the workers. Therefore, organizations should avoid stereotyping and emphasize the promotion of the bottom-up approach, in the form of humble leadership, to promote teamwork and creative performance. Organizations should focus on creating synergy by promoting new ideas and innovative behaviors and, in response, listen and accept subordinates’ advice for the improvement of the working environment and the coordination of work.

As our study shows, humble leaders accepts their mistakes and are ready to receive advice from team members to improve themselves. This kind of leadership in an organization also acknowledges the limitations of employees, which further promotes a conducive working environment. This leadership style can improve the chances of survival in a competitive market.

All leaders should focus on their strengths and minimize their weaknesses to strengthen their leadership. Leaders should capitalize on their strengths and deal with their weaknesses in parallel to obtain the maximum out of their leadership qualities. One of the root causes of conflict is the communication gap between employees and their immediate boss. To avoid any misunderstandings, this gap should be filled, as it ultimately decreases the performance and productivity of an organization, which may also affect the use of resources in the organization.

Finally, leaders should be proactive in their approach and be able to sense negative vibes. This will further help leaders to overcome conflict before it really happens. Leaders should develop problem-solving skills so that relationship conflict can be avoided, and utilize these skills to prevent every opportunity for misunderstanding among the employees in an organization.

### 5.3. Limitations and Future Study

Like any research, this study also has a few limitations. Firstly, the analysis of the data provides insufficient evidence for assessing causation. To address this, future studies could be conducted on the topic of employee conflict over time by analyzing the time lag and longitudinal data. Secondly, the measures of creative performance used in this study are not entirely objective. While both objective and subjective measures of creative performance have been found to be related, adding objective measures of creative performance would provide more comprehensive and concrete results. Future studies could consider using corporate performance measures to avoid subjectivity.

Data were collected and analyzed from only two cities in Pakistan, and specifically from the telecom sector only. To address this limitation, future studies could expand the scope of this study by including more regions and also the manufacturing sector. This would help to enhance the generalizability of the findings. Lastly, the lack of participation by female workers and their not having a leadership role is another noted limitation of this study. Future studies could aim for gender balance and assess whether there are any differences in creative performance and employee conflict when the leadership role is interchanged.

## Figures and Tables

**Figure 1 behavsci-13-00483-f001:**
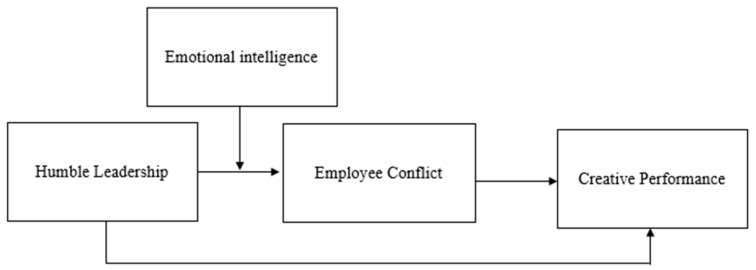
Proposed Research Model.

**Figure 2 behavsci-13-00483-f002:**
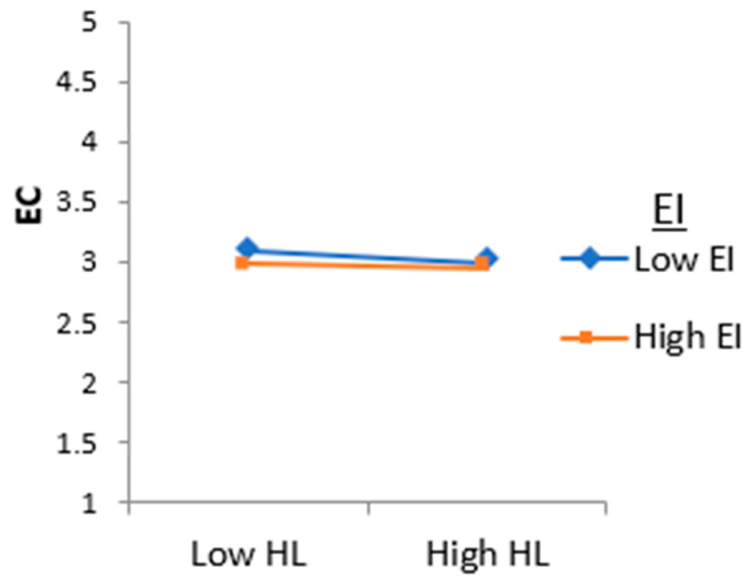
Moderating effect of emotional intelligence on the relationship between humble leadership and employee conflict.

**Figure 3 behavsci-13-00483-f003:**
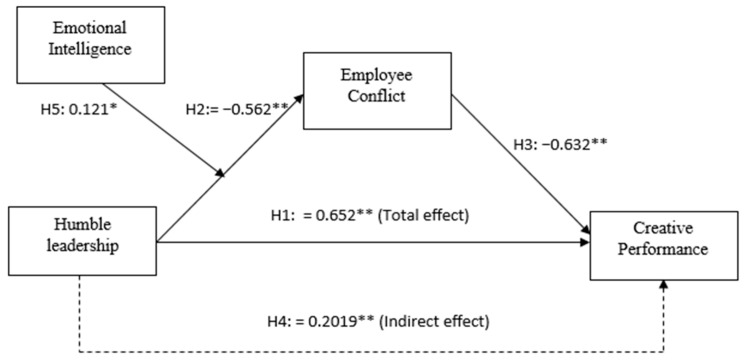
Beta coefficients of total, direct and indirect effects of HL, CP, EC and EI at * *p* < 0.05, ** *p* < 0.01.

**Table 1 behavsci-13-00483-t001:** Confirmatory factor analysis.

Constructs	Items	Loadings	Cronbach’s Alpha	Constructs	Items	Loadings	Cronbach’s Alpha
Humble leadership	HL1	0.78	0.968	Employee conflict	EC1	0.78	0.956
	HL2	0.86			EC2	0.82	
	HL3	0.85			EC3	0.91	
	HL4	0.91			EC4	0.9	
	HL5	0.88			EC5	0.91	
	HL6	0.88			EC6	0.90	
	HL7	0.92			EC7	0.84	
	HL8	0.88		Emotional intelligence	EI1	0.88	0.984
	HL9	0.85			EI2	0.90	
Creative performance	CP1	0.92	0.976		EI3	0.88	
	CP2	0.93			EI4	0.90	
	CP3	0.95			EI5	0.88	
	CP4	0.94			EI6	0.88	
	CP5	0.91			EI7	0.86	
	CP6	0.88			EI8	0.89	
	CP7	0.86			EI9	0.88	
	CP8	0.84			EI10	0.90	
	CP9	0.82			EI11	0.90	
					EI12	0.90	
					EI13	0.91	
					EI14	0.89	
					EI15	0.89	
					EI16	0.87	

Note: All factor loadings are significant at the *p* < 0.001 level.

**Table 2 behavsci-13-00483-t002:** Validity, reliability, descriptive statistics, and correlations.

	CR	AVE	MSV	MaxR(H)	HL	CP	EC	EI
HL	0.965	0.754	0.450	0.968	0.868			0.395 **
CP	0.973	0.801	0.450	0.978	0.671 **	0.895		0.416 **
EC	0.955	0.751	0.359	0.960	−0.599 **	−0.597 **	0.867	−0.342 **
EI	0.984	0.790	0.173	0.984				0.889
Mean					3.23	3.19	3.18	3.29
SD					1.17	1.1816	1.13	1.21

Note: CR—Composite reliability; AVE—Average variance extracted; MSV—maximum shared variance; MaxR—Maximum reliability. The square roots of average variance extracted (AVE) are given diagonally for each variable. ** = *p* < 0.01

**Table 3 behavsci-13-00483-t003:** Measurement model fit.

Model	CMIN (ꭓ^2^)	df	ꭓ^2^/df	SRMR	GFI	RMSEA	NFI	TLI (NNFI)	CFI
Model fit	2001.04	727	2.75	0.072	0.765	0.074	0.905	0.929	0.937

**Table 4 behavsci-13-00483-t004:** Regression analysis.

Relationship	β	t	Sig.	R^2^	F	Sig.	Hypotheses
H1: HL => CP	0.652	15.257	0.000 ***	0.421	232.789	0.000 ***	Supported
H2: HL => EC	−0.562	−12.893	0.000 ***	0.342	166.22	0.000 ***	Supported
H3: EC => CP	−0.632	−13.619	0.000 ***	0.367	185.48	0.000 ***	Supported

Note: HL—Humble leadership; CP—Creative performance; EC—Employee conflict. *** *p* < 0.001.

**Table 5 behavsci-13-00483-t005:** Mediation analysis.

Relationship	DE	IE	TE	LLCI	ULCI	Mediation
H4: HL => EC => CP	0.4497	0.2019	0.6517	0.1422	0.2817	Partial

Note: DE—Direct effect; IE—Indirect effect; TE—Total effect; LLCI—Lower-level confidence level; ULCI—Upper-level confidence level.

**Table 6 behavsci-13-00483-t006:** Direct and moderated effect of humble leadership on employee conflict.

	Model 1			Model 2		
Variables	B	β	SE	B	β	SE
Age	0.032	0.040	0.057	0.049	0.061	0.072
Gender	−0.058	−0.015	0.138	−0.037	−0.010	0.174
Experience	0.011	0.010	0.076	−0.023	−0.022	0.095
Industry size	−0.044	−0.039	0.042	−0.013	0.053	0.053
Constant	3.182 ***		0.050	3.135 ***		0.054
Humble leadership	−0.506 ***	−0.527 ***	0.046	−0.465 ***	−0.483 ***	0.049
Emotional intelligence	−0.148 **	−0.159 ***	0.045	−0.136 **	−0.147 **	0.044
HL × EI (Interaction)				0.091 *	0.121 *	0.037
R^2^	0.364				0.376	
∆R^2^ F	91.212 ***				0.012	63.860 ***

Note: N = 322; * *p* < 0.05; ** *p* < 0.01; *** *p* < 0.001.

**Table 7 behavsci-13-00483-t007:** Moderated mediation effect of emotional intelligence.

Mediator	Moderator	Conditional Indirect Effect				Moderated Mediation Effect			
		Indirect Effect	SE	95% Confidence Interval		INDEX	SE	95% Confidence Interval	
				LLCI	ULCI			LLCI	ULCI
Employee conflict	Low	0.2184	0.0469	0.1316	0.3174	−0.0328	0.0163	−0.0688	−0.0048
	Medium	0.1481	0.0285	0.0954	0.2068				
	High	0.1276	0.0295	0.0736	0.1892				

Note: Conditional indirect effects of emotional intelligence: ULCI—upper level of 95% confidence interval; LLCI—lower level of 95% confidence interval.

## Data Availability

Data are available on request from the corresponding author.

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
