# Peer review of "Creative Performance and Conflict through the Lens of Humble Leadership: Testing a Moderated Mediation Model"

_behavsci, 2023, doi:10.3390/bs13060483_

Round 1

Reviewer 1 Report

The paper under review is based on a good literature review and the research has been adequately designed and conducted. There are suggestion to be considered.

To be precise, when referring to the sample size and characteristics in page 6, there is a mention to the number of initially intended surveys, the number of finally obtained questionnaires, and the characteristics of the sample individuals. However, there is no mention to the representativeness of such sample regarding the total population (confidence level and sampling error) or in terms of the mentioned characteristics. In a similar sense, there is no mention to the concrete time when data were obtained. Both comments would be advisable, as providing an useful information to properly contextualize the research.

Regarding Figure 1, it would be better placed after and not before the definition of the three hypotheses. What is more relevant, the relationships (arrows) appearing in the figure should be according to what is indicated i the hypotheses (at the moment, the hypotheses are not reflected in the figure). Later on, it would be advisable to include again the figure with the values obtained in order to provide a clear indication of the hypotheses acceptance/refutation. A further comment on this issue (hypotheses acceptance/reputation) would be welcome in the section on "discussion" (which, according to its content, could be renamed as "conclusions").

The quality of English is quite good. Only a minor review by an English native/expert would be advised.

Author Response

Response to Reviewer 1 Comments

Point 1: To be precise, when referring to the sample size and characteristics in page 6, there is a mention to the number of initially intended surveys, the number of finally obtained questionnaires, and the characteristics of the sample individuals. However, there is no mention to the representativeness of such sample regarding the total population (confidence level and sampling error) or in terms of the mentioned characteristics. In a similar sense, there is no mention to the concrete time when data were obtained. Both comments would be advisable, as providing an useful information to properly contextualize the research.

Response 1: The comments were incorporated in page number 6 from line number 269 to line number 279 as follows.

Since we used convenient sampling, specific steps were taken to improve the representativeness of the sample. To minimize selection bias, we made efforts to select participants from various accessible locations and sources so that a diverse range of perspectives could be provided. It was ensured that a wide range of opinions and experiences were provided that represent the population as a whole. Moreover, we ensured the heterogeneity of the sample by rigorously seeking participants having various demographic characteristics such as age, gender, tenure, and socio-economic status etc. Finally, we applied rigorous data analysis techniques to and carefully interpreted the results to reduce any biases due to convenient sampling. The data was collected in a 6 months’ time frame from October 2021 to March 2022.

Point 2: Regarding Figure 1, it would be better placed after and not before the definition of the three hypotheses. What is more relevant, the relationships (arrows) appearing in the figure should be according to what is indicated i the hypotheses (at the moment, the hypotheses are not reflected in the figure). Later on, it would be advisable to include again the figure with the values obtained in order to provide a clear indication of the hypotheses acceptance/refutation. A further comment on this issue (hypotheses acceptance/reputation) would be welcome in the section on "discussion" (which, according to its content, could be renamed as "conclusions").

Response 2: The comments were incorporated in page 11 from line 414 to 418. We included the figure again and mentioned the results of our accepted hypotheses in the form of beta coefficients.

Further comments are added in the discussion section from line 446 to 448 about the findings of the study. In line 444 we included the word “Conclusion” too.

NOTE: Your valuable advice about the quality of English is also incorporated. We send our menu script to an English native to eliminate any chance of error in English language. He only made a minor change which is mentioned in line 408.

Reviewer 2 Report

Dear Authors

I found your work to be interesting and I thoroughly enjoyed reading it. In this respect I am sure that it will be attractive for readers of the journal.

Just a few minor corrections suggested;

1. In the Introduction section (line 30 of your work) the authors present humble leadership as HI. It is necessary to present in full and then use the shorted version (humble leadership (HL)).

2. In the Methods section (line 254) the authors mention their sample consisted to employees working in the telecom sector. Even though it is briefly mentioned in the first few lines of the Theoretical Implications/ Discussion section, justification needs to be provided for the choice of this sector in the Methods section.

3. Also in the Discussions section (line 418) the authors mentioned the data was collected from employees from two cities. Again, justification for the choice of two cities and which two cities should also be provided in the Methods section.

4. In the Sample and Date Collection / Methods section (line 271) the authors mentioned that 500 surveys were distributed. Why 500? What was the population? The selected sample size needs to be justified.

Author Response

Response to Reviewer 2 Comments

Point 1:  In the Introduction section (line 30 of your work) the authors present humble leadership as HI. It is necessary to present in full and then use the shorted version (humble leadership (HL)).

Response 1:

Responding to your valuable suggestion regarding line 30. We incorporated it. First we wrote the full form then its short version. According this suggestion we also incorporated this suggestion in line 40 and 52

Point 2: In the Methods section (line 254) the authors mention their sample consisted to employees working in the telecom sector. Even though it is briefly mentioned in the first few lines of the Theoretical Implications/ Discussion section, justification needs to be provided for the choice of this sector in the Methods section.

Response 2:

We incorporated suggestion 2 by respected reviewer from line 254 to 259.

We selected the telecom sector because of the rapid technological changes in this sector. Due to these changes the employees are on their toes to incorporate these changes to meet the customer’s demands. This ever changing working environment requires creativity and doing the things in innovative and creative way. Moreover, another reason for the selection of this sector was the lack of research in Pakistani as well as in international context.

Point 3: Also in the Discussions section (line 418) the authors mentioned the data was collected from employees from two cities. Again, justification for the choice of two cities and which two cities should also be provided in the Methods section.

Response 3: The suggestion is incorporated from line 259 to 262

The data was collected from two provincial capitals of Pakistan namely Quetta and Karachi. We collected data from these cities to increase the generalizability of our study. Because both the cities are different in culture, language and working environment.

Point 4: In the Sample and Date Collection / Methods section (line 271) the authors mentioned that 500 surveys were distributed. Why 500? What was the population? The selected sample size needs to be justified.

Respose 4:  The original line (271) is now changed to line 295. Because in line 271 we incorporated another sugesstion from reviewer 1. Therefore; Response to the suggestion 4 is addressed from line 295 to 300.

 Haier et. al (2012) Sets a criteria for the minimum sample size. According to this criteria, there should be five or less than five variables in a study and there measurement items should be at least three. If a study meets the above mentioned threshold then the minimum sample size should be 100. Moreover according to this criteria the communalities sholud be greater or equal to 0.6. Our sample size, number of variables and communalities fulfills the required standards to justify our sample size.